# IdealGPT: Iteratively Decomposing Vision and Language Reasoning via Large Language Models

**Haoxuan You[1]\*, Rui Sun[1]\*, Zhecan Wang[1]\*, Long Chen[2], Gengyu Wang[1]**
**Hammad A. Ayyubi[1], Kai-Wei Chang[3], Shih-Fu Chang[1]**

[1] Columbia University    [2] HKUST    [3] University of California, Los Angeles

{hy2612, rs4110, zw2627, gengyu.wang, ha2578, sc250}@columbia.edu

longchen@ust.hk, kwchang@cs.ucla.edu

## Abstract

The field of vision-and-language (VL) understanding has made unprecedented progress with end-to-end large pre-trained VL models (VLMs). However, they still fall short in zero-shot reasoning tasks that require *multi-step* inferencing. To achieve this goal, previous works resort to a divide-and-conquer pipeline. In this paper, we argue that previous efforts have several inherent shortcomings: 1) They rely on domain-specific sub-question decomposing models. 2) They force models to predict the final answer even if the sub-questions or sub-answers provide insufficient information. We address these limitations via **IdealGPT**, a framework that *iteratively* decomposes VL reasoning using large language models (LLMs). Specifically, IdealGPT utilizes an LLM to generate sub-questions, a VLM to provide corresponding sub-answers, and another LLM to reason to achieve the final answer. These three modules perform the divide-and-conquer procedure iteratively until the model is confident about the final answer to the main question. We evaluate IdealGPT on multiple challenging VL reasoning tasks under a zero-shot setting. In particular, our IdealGPT outperforms the best existing GPT-4-like models by an absolute 10% on VCR and 15% on SNLI-VE. Code is available at https://github.com/Hxyou/IdealGPT.

## 1 Introduction

The field of vision-and-language (VL) understanding has witnessed a proliferation of pre-trained VL models (VLMs) (Yu et al., 2022; You et al., 2023; Alayrac et al., 2022; Zhu et al., 2023b; Liu et al., 2023). They are usually pre-trained and fine-tuned in an end-to-end fashion, *i.e.*, these models always directly make final predictions in a single step. With abundant pre-trained knowledge, they already achieve impressive results in comparison to human performance across many downstream

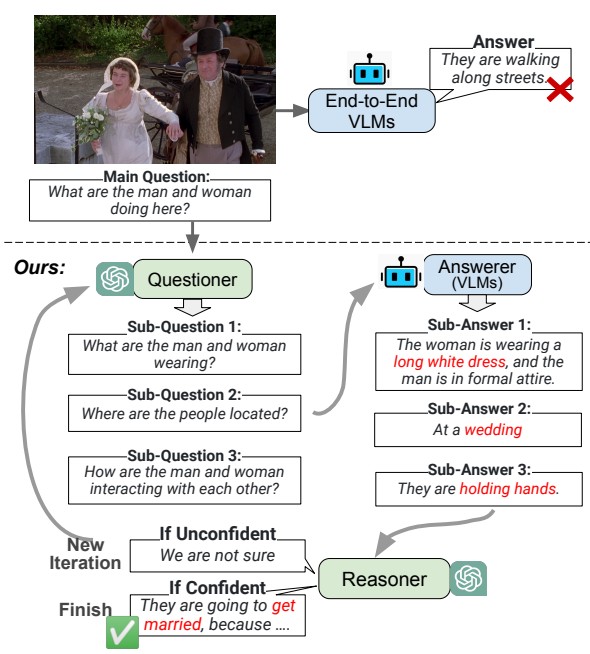

Figure 1: Comparisons between the pipelines of prevalent end-to-end VLMs (Upper) and our proposed IdealGPT (Below) for VL reasoning tasks.

VL tasks. However, they still struggle to address zero-shot VL reasoning tasks that require intricate or multi-step inferencing such as visual commonsense reasoning (VCR) (Zellers et al., 2019), as exemplified in Figure 1. Despite the overall success of these models, the difficulties with zero-shot reasoning settings represent a serious challenge in the current state of VL research.

In comparison, humans are able to excel in these tasks effortlessly. As in Figure 1, to answer the question, *"What are the man and woman doing here?"*, we humans would intuitively approach it in a divide-and-conquer fashion by starting with simple questions like "What are they dressed as?" to identify people in the image first. Then we may advance further to wonder about their interactions, "How do they interact?" and the location, "Where

---

\* Equal Contribution

are they at?". Even though it may seem challenging to answer the original question directly, it is much easier to answer the three sub-questions. After answering the three simpler sub-questions, we can understand the image comprehensively, and utilize commonsense knowledge to obtain a conclusion: "*they are going to get married*". This step-by-step procedure can be explicitly formulated as a decompositional reasoning process of three key steps: (1) dividing the main question into multiple sub-questions focusing on visual details, (2) answering easier sub-questions, and (3) reasoning upon the sub-answers to address the main question.

Inspired by this divide-and-conquer manner, several works solve VL reasoning with this compositional pipeline: collecting the sub-questions dataset (Selvaraju et al., 2020), generating sub-questions for the main question (Uehara et al., 2022; Wang et al., 2022b,d), and utilizing sub-questions to help to answer the main question (Selvaraju et al., 2020; Uehara et al., 2022; Wang et al., 2022b). Nonetheless, several drawbacks still exist and hinder their practice: 1) Existing methods rely on task-specific trained models to generate sub-questions, which are not generalizable. 2) Existing methods impractically assume that the sub-questions and sub-answers generated in one round can guarantee sufficient evidence to address the main question, which is profoundly incorrect in practice. For instance, the generated sub-questions might be not informative enough or deviate from the main question. These predicted sub-answers may be noisy and raise conflicts due to possible misprediction. Therefore, existing methods may cause irrational final predictions or be forced to learn spurious bias to guess the final answer.

To address these above-mentioned issues, we proposed a new framework **IdealGPT**, which **I**teratively **de**composes vision **a**nd **l**anguage reasoning with large language models (LLMs). Specifically, IdealGPT employs two LLMs (*e.g.*, GPT (Ouyang et al., 2022; OpenAI, 2023) in our experiments) as the `Questioner` and the `Reasoner`, and a pretrained VLM as the `Answerer`. In this framework, these three agents interact with each other to perform the divide-and-conquer procedure iteratively until finding a confident answer to the main question. As shown in Figure 1, the `Questioner` first raises sub-questions decomposed from the main question, and the `Answerer` subsequently replies with the corresponding sub-answers. Subse-

quently, the `Reasoner` analyzes cumulative information extracted from sub-answers to infer the possible answer to the main question. If the `Reasoner` ascertains that the evidence gathered so far is insufficient to confidently answer the main question (either due to uninformative sub-questions or noisy sub-answers), it loops back to the `Questioner` with its analysis of the gathered evidence. Hence, `Questioner` would purposely try to generate more targeted sub-questions to obtain more informative evidence. These iterations of the QA loop would continue to be initiated until `Reasoner` is confident of resolving the main question or the number of iterations reaches a predefined threshold.

Compared with previous compositional and end-to-end methods, the proposed IdealGPT brings several significant benefits: 1) **Transparency and Interpretability**. It is straightforward to pinpoint which sub-answer or reasoning step results in the undesired final answer. Meanwhile, multi-round interactions allow models to showcase their understanding and reasoning process step by step which leads to the final answer. 2) **Modularity**. With the rapid development of LLMs and VLMs, `Questioner/Reasoner` and `Answerer` can easily be updated to the more powerful LLM and VLM to improve performance. 3) **Robustness**. Existing models still heavily suffer from problems like superficial biases, inconsistent predictions, or hallucination. All of these can lead to conflicted and noisy evidence during reasoning steps. Our multi-rounds can robustly consider models' both noisy and accurate predictions to converge to the most confident answer. 4) **Generalizability**. IdealGPT can be seamlessly applied to multiple tasks. This is because of the fact that various VL tasks require reasoning skills and can be inherently formatted as question-answer tasks. Moreover, IdealGPT illustrates strong zero-shot ability with no training or finetuning on specific tasks.

We quantitatively evaluate IdealGPT on several challenging VL reasoning tasks in a zero-shot setting, including VCR and Visual Entailment (SNLI-VE) (Xie et al., 2019). Since zero-shot VCR and SNLI-VE are too challenging for most of the previous VLMs (Li et al., 2023; Yu et al., 2022; Alayrac et al., 2022), we found only GPT-4-like models based on instruction-tuned LLMs, such as MiniGPT4 (Zhu et al., 2023b) and LLaVA (Liu et al., 2023), are capable of tackling the tasks. Compared with the above-mentioned GPT-4-like mod-

els, IdealGPT outperforms the best by an absolute 10% in VCR and 15% in SNLI-VE.

## 2 Related Works

### 2.1 Compositional QA in Vision/Language

Answering multi-hop reasoning questions directly can be challenging in NLP. Press et al. (2022) investigate the ability of language models to perform compositional reasoning tasks where the final solution depends on correctly composing the answers. Wang et al. (2022c) exploit a self-consistency method to sample a diverse set of reasoning paths and then filter and aggregate by choosing the most consistent answer. Yoran et al. (2023) sample multiple reasoning chains and mix information between them to select the most relevant facts in generating an explanation and predicting the answer. In VQA, in order to investigate the reasoning process and promote the reliability of models, SQuINT (Selvaraju et al., 2020) collect VQA-introspect dataset providing low-level perception sub-questions to answer the complex reasoning questions. Some methods (Uehara et al., 2022; Wang et al., 2022b) decompose the original complicated questions into several informative sub-questions. By answering these sub-questions, they can help to answer the original questions. These existing methods rely on task-specific trained models and generate sub-questions and sub-answers in one round, which prevents their generalizability and reasoning ability in practical problems. However, IdealGPT can be seamlessly utilized in different tasks by slightly adjusting the prompt. Moreover, our iterative approach can efficiently solve challenging tasks such as VCR and SNLI-VE without further training.

### 2.2 End-to-End Vision-Language Models

VL pre-training models (Li et al., 2019; Chen et al., 2020; Li et al., 2022; You et al., 2022; Yu et al., 2022) are pre-trained on large-scale image-text pairs, which enable these models' joint understanding between different modalities. Recently, there is a trend to utilize the knowledge from LLMs and align visual features to the text space. Flamingo (Alayrac et al., 2022) inserts cross-attention layers into LLMs to import visual features and employs billions of image-text pairs to pre-train the new layers. BLIP-2 (Li et al., 2023) is powered by the pre-trained visual encoder and LLMs. It uses a lightweight Querying Transformer (Q-Former) following a two-stage pre-training to bridge the

modality gap. Inspired by InstructGPT (Ouyang et al., 2022), to improve the generalization performance of LLMs to unseen tasks and align users' intentions, some VL models finetune the pre-trained models using extra instruction-tuning data. LLaVA (Liu et al., 2023) leverages the trainable projection layer to project the output from the visual encoder to the LLM and utilizes VL conversational data generated by GPT-4 (OpenAI, 2023) to finetune the LLM. MiniGPT4 (Zhu et al., 2023b) employs the pre-trained visual encoder and Q-Former from BLIP-2 and uses image captions generated by Chat-GPT to perform training on the LLM and the single linear projection layer.

### 2.3 GPT-Aided Visual Reasoning

LLMs are pre-trained on colossal corpus so they can attain rich prior knowledge and strong reasoning ability. Recently, a trend has emerged that leverages LLMs in combination with a range of vision or multimodal models. These approaches create a system capable of addressing various multimodal tasks without the need for additional training. MM-React (Yang et al., 2023), HuggingGPT (Shen et al., 2023), Chameleon (Lu et al., 2023), Visual Chat-GPT (Wu et al., 2023) regard GPT (*i.e.*, ChatGPT, GPT-4) as a controller to coordinate and collaborate with other models (*e.g.*, visual foundation models) to tackle complicated multimodal tasks. VisProg (Gupta and Kembhavi, 2022) and ViperGPT (Surís et al., 2023) exploit GPT-3 (Brown et al., 2020) and CodeX (Chen et al., 2021) to generate a program to solve VL tasks in a one-round query answering. ChatCaptioner (Zhu et al., 2023a) lets ChatGPT and BLIP-2 interact to accomplish image captioning in a dialogue approach. All of them borrow the strong reasoning ability from LLMs and boost performance in a wide range of VL tasks. Different from ViperGPT, VisProg, and ChatCaptioner, IdealGPT solves VL tasks iteratively in a multi-round manner and our proposed method can be conveniently deployed in a diverse set of VL tasks such as VCR and SNLI-VE. More details are in Sec. 3.4.

## 3 Method

In this section, we introduce the proposed Ideal-GPT. Our focus is on the tasks of open-domain VQA, where a question $q$ is asked about an image $I$. There are three components in IdealGPT framework: a `Questioner`, an `Answerer`, and a `Reasoner`. In each iteration, based on $q$ and

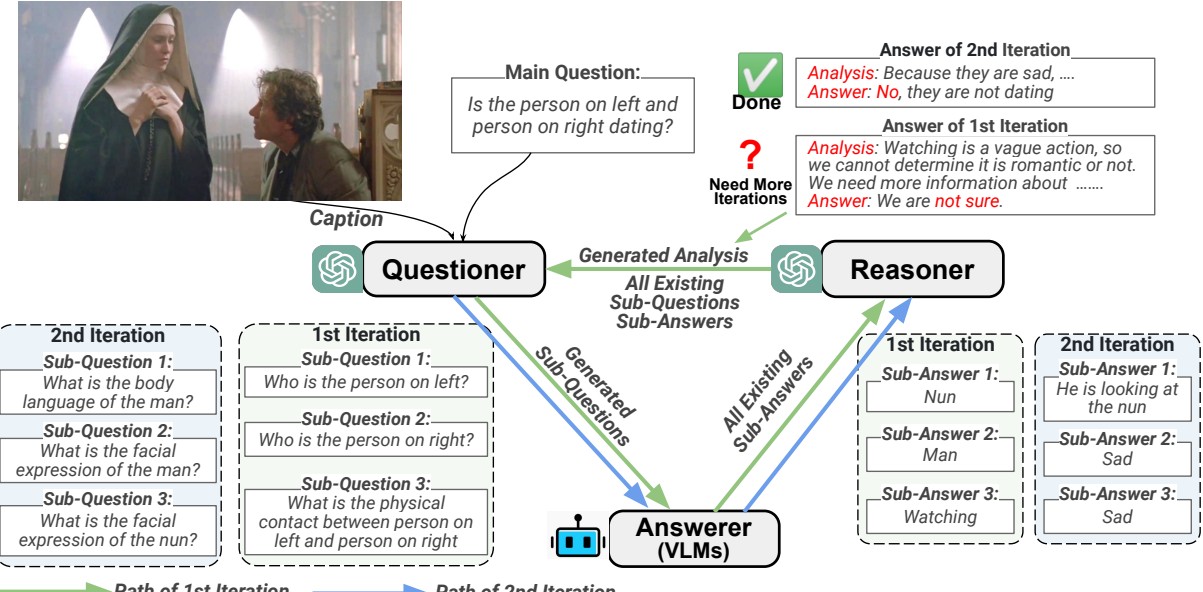

Figure 2: The pipeline of proposed IdealGPT. We use an example in VCR validation set for illustration, which is finished in 2 iterations. In the initial iteration, the main question is decomposed into multiple sub-questions, which are answered by Answerer, and then the Reasoner summarizes generated sub-answers to decide whether a confident answer can be deduced. Since Reasoner is not sure about the answer in the 1st iteration, it generates the analysis, and all existing information is input into Questioner again for another new iteration. The iterative process ends when Reasoner is confident about an answer or the maximum number of iterations is reached.

$I$, Questioner first raises a set of sub-questions $SubQ = \{sq_1, ..., sq_i\}$ (Sec. 3.1), then Answerer generates the corresponding sub-answers $SubA = \{sa_1, ..., sa_i\}$ (Sec. 3.2), and Reasoner analyzes both $SubA$ and $SubQ$ to decide if a confident answer $a$ to the main question $q$ can be derived (Sec. 3.3). If a confident answer cannot be inferred in the current iteration, Questioner is prompted to ask additional supplementary sub-questions, and another "Questioner-Answerer-Reasoner" iteration is triggered. This loop keeps iterating until the Reasoner finds a confident final answer or the number of iterations reaches a predefined maximum. The overall pipeline is shown in Figure 2.

## 3.1 Questioner

In previous works (Uehara et al., 2022; Wang et al., 2022b), sub-questions are generated by models trained on specific sub-questions data (Selvaraju et al., 2020). However, since the training data are specifically annotated for the samples in Antol et al. (2015), these sub-question generators cannot scale and generalize to different domains and complex reasoning tasks. Annotating a sub-question dataset covering all types of visual reasoning tasks and image domains is also infeasible. Recently, LLMs (Ouyang et al., 2022; OpenAI, 2023; Anil et al.,

2023) have demonstrated a strong ability to follow instructions and reason with human knowledge. Additionally, some works (Surís et al., 2023; Wu et al., 2023; Zhu et al., 2023a) have utilized LLMs to aid visual tasks, which demonstrates that LLMs have acquired diverse visual knowledge to a certain degree as well. Inspired by these findings, we prompt GPT as a Questioner to raise sub-questions. By default, ChatGPT (Ouyang et al., 2022) is used. While GPT-4 (OpenAI, 2023) is a stronger alternative, it is costlier.

The input in VQA tasks usually includes a main question $q$ and an image $I$, and sometimes answer candidates $A = \{a_1, ..., a_n\}$ if the task is formatted as a multiple-choice QA problem. The target of the Questioner is first to recognize the evidence needed to address $q$ and then decompose it into sub-evidences. To acquire those sub-evidences, Questioner would then generate a set of sub-questions $SubQ = \{sq_1, ..., sq_i\}$. For achieving quality results, we also design a prompt $P_q$ as an instruction for GPT to understand the objective and the desired output. For each task, the prompt is slightly different to accommodate the task descriptions[1]. With solely the main question $q$ and prompt $P_q$ input into the Questioner, we empiri-

---

[1] See the detailed prompts in Appendix D

cally found that the generated sub-questions from initial iterations tend to be too generic and uninformative. This could be because LLMs don't see the images and as such are devoid of the visual input. To facilitate the Questioner to understand the image and generate more informative questions, we provide a short caption $C_I$ generated by a VLM as an additional input to the Questioner. Therefore, in the first iteration, the sub-question generation can be formulated as follows:

$$SubQ_1 = \text{ChatGPT}(q, C_I, P_q).$$

As mentioned before, there may not be sufficient evidence for the Reasoner to address the main question after only one iteration. This can be due to common issues like the sub-questions are not informative enough or conflict/noise existing among sub-answers. In this case, IdealGPT would prompt the Reasoner to generate an explanation/analysis regarding why it may not have sufficient evidence to address the main question. Subsequently, we would loop back to the Questioner to generate additional supplementary sub-questions. In the $t$-th iteration ($t > 1$), Questioner accepts all previous sub-questions $SubQ_{1:t-1}$ and sub-answers $SubA_{1:t-1}$, and the previous analysis $E_{t-1}$ from Reasoner (c.f., Sec. 3.3) as additional input:

$$
\begin{aligned}
SubQ_t = &\text{ChatGPT}(q, C_I, P_q, \\
&SubQ_{1:t-1}, SubA_{1:t-1}, E_{t-1}),
\end{aligned}
$$

where $SubQ_{1:t-1} = \{SubQ_1 \cup ... \cup SubQ_{t-1}\}$ and $SubA_{1:t-1} = \{SubA_1 \cup ... \cup SubA_{t-1}\}$. Previous sub-questions and sub-answers can inform Questioner what has been asked and solved, and the analysis can guide Questioner to generate more specific sub-questions, such as sub-questions to collect more informative evidence about a specific visual object and so on.

## 3.2 Answerer

Given the generated sub-questions $SubQ$, the goal of Answerer is to answer them correspondingly to provide evidence for answering the main question. In IdealGPT, the Answerer is a pre-trained VLM without finetuning on any dataset to keep intact its generalization ability. Each sub-question is answered separately:

$$sa_i = \text{VLM}(sq_i, I),$$

where $sq_i \in SubQ$ and $sa_i \in SubA$. It is noted that theoretically, the Answerer can not only be

end-to-end VLMs but also VL systems such as Surís et al. (2023); Gupta and Kembhavi (2022); Shen et al. (2023).

## 3.3 Reasoner

GPT-like LLMs (Ouyang et al., 2022; OpenAI, 2023; Anil et al., 2023) have shown impressive summarization and reasoning ability with commonsense knowledge. Therefore, like Questioner, we also choose ChatGPT as the Reasoner but prompt it differently. Specifically, the input to Reasoner contains main question $q$, caption $C_I$, all existing sub-questions $SubQ_{1:t}$ and corresponding sub-answers $SubA_{1:t}$. And the Reasoner is prompted to generate both the analysis and the final answer with its prompt $P_R$[1]:

$$E_t, a = \text{ChatGPT}(SubQ_{1:t}, SubA_{1:t}, q, C_I, P_R).$$

If the Reasoner is not confident about the final answer, it is instructed to faithfully indicate that by generating a specific response such as "We are not sure". If this particular response is detected, we start another iteration by asking the Questioner to add supplementary sub-questions. The above procedure forms a loop among the three agents, which will stop if the Reasoner can find a confident answer or the number of iterations reaches a predefined bound (a hyperparameter).

## 3.4 Comparison with Other Methods

***v.s.*ViperGPT/VisProg**. VisProg (Gupta and Kembhavi, 2022) and ViperGPT (Surís et al., 2023) utilize LLMs to decompose VL tasks into steps of explicit conditional logic in coding based on low-level visual-spatial detection. IdealGPT shares a similar idea of decomposition or divide-and-conquer. However, IdealGPT can iteratively go through the divide-and-conquer process until collecting sufficient evidence to generate a confident answer. This multi-pass process involves the continuous refinement of the set of sub-questions and even a re-correctifying mechanism to the final answer prediction. Conversely, ViperGPT/VisProg performs programs in one pass regardless of whether the predicted answer is confident or not. This difference also applies to Yang et al. (2023); Shen et al. (2023); Lu et al. (2023). Moreover, ViperGPT is limited by the set of available APIs, VisProg is limited by the set of available commands, and Neurosymbolic VQA is limited by the set of primitive operations in the Domain Specific Language.

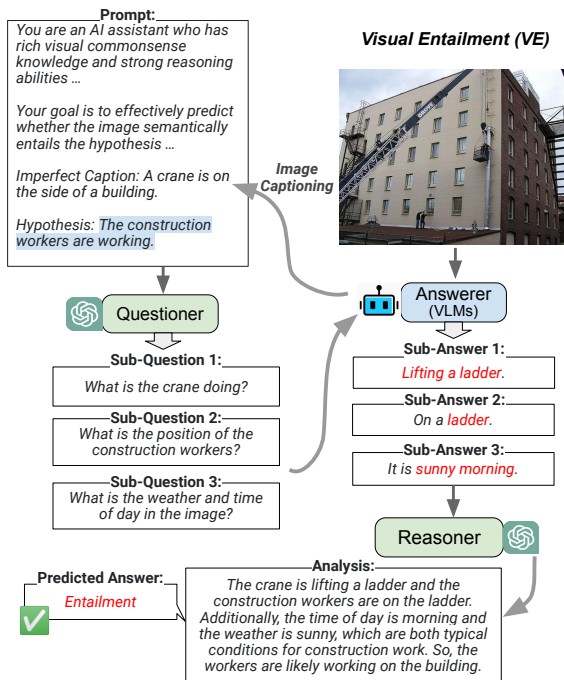

Figure 3: The illustration of how proposed IdealGPT works in SNLI-VE. IdealGPT exhibits its perceptional ability in sub-question 1 and 2. Plus, it shows a strong commonsense reasoning ability in sub-question 3.

**v.s.ChatCaptioner.** ChatCaptioner (Zhu et al., 2023a) lets ChatGPT and BLIP-2 interact with each other to caption images in a dialogue system. IdealGPT shares the idea of utilizing ChatGPT to generate questions and VLMs to answer them. But IdealGPT focuses on the VL reasoning tasks and incorporates the iterative design for generalized zero-shot vision language reasoning.

## 4 Experiments

In this section, we evaluate our method quantitatively on two main tasks - Visual Commonsense Reasoning (VCR) and Visual Entailment (SNLI-VE). We introduce datasets and models first. Then, we show superior zero-shot performance on VCR and SNLI-VE compared to other existing models. Afterward, ablation studies are conducted to confirm the effectiveness of our proposed method, with error analysis for a deep dive into our method. In addition to VCR and SNLI-VE, we also showcase the experimental results of Augmented Outside Knowledge Visual Question Answering (A-OKVQA) (Schwenk et al., 2022). Since it is more of a knowledge-based task requiring less complex reasoning ability, we put it in Appendix C.

### 4.1 Experimental Setup

**Datasets.** In this paper, we employ two VL datasets, VCR (Zellers et al., 2019) and SNLI-VE (Xie et al., 2019) as they represent the two typical VL reasoning tasks, visual question answering and visual entailment. Different from the traditional VQA (Antol et al., 2015) task, VCR needs commonsense knowledge to understand the visual scenes and requires multiple-steps reasoning to answer the question. Also, in terms of the task format, it is a multiple-choice QA problem. SNLI-VE originates from Stanford Natural Language Inference (SNLI) (Bowman et al., 2015), a text entailment (TE) task based on Flicker30k (Young et al., 2014). It extends TE into the visual domain and asks the model whether the image is semantically entailed/neutral/contradicted to the text hypothesis, thus it can be treated as a three-category classification task. In VCR and SNLI-VE, we randomly select 5000 samples from the val/dev split of the dataset and evaluate our method in the **zero-shot** scenario with the accuracy for evaluation.

**Models.** We choose ChatGPT to act as the Reasoner and Questioner and access it via "gpt-3.5-turbo" API. It should be noted that we set the temperature as 0 to reduce the randomness so that we can have a stable result to see how our proposed method performs in different tasks. As for the Answerer, three pre-trained VLMs (BLIP-2, MiniGPT4, and LLaVA) are selected to serve for a comprehensive comparison. It's noted that all VLMs we use are pre-trained-only without being finetuned on any dataset to guarantee zero-shot generalizability. We design three simple prompts for these models respectively (more details can be found in the appendix D) to help them better answer sub-questions. Further, in practice, these VLMs are also used to produce image captions so that the ChatGPT can reserve a general understanding of the unseen image initially.

### 4.2 Visual Commonsense Reasoning

VCR covers various commonsense in diverse visual scenes, including temporal, spatial, causal, and mental commonsense, etc. It's formatted as a multiple-choice answering problem, and to find the correct answer, the model is expected to reason among four answer candidates to find the correct answer. In VCR, it often happens that there are multiple persons in one image, and if the question mentions one of them, the bounding box will be

|  |  | Acc.(%) |
|---|---|---|
|  | Random Guess | 25 |
| Sup. | R2C (Zellers et al., 2019) | 63.8 |
| Sup. | VisualBERT (Li et al., 2019) | 70.8 |
| Sup. | MerlotReserve (Zellers et al., 2022) | 84.0 |
| ZS. | BLIP-2 (Li et al., 2023) | - |
| ZS. | MiniGPT4 (Zhu et al., 2023b) | 40.6 |
| ZS. | LLaVA (Liu et al., 2023) | 28.3 |
| ZS. | **IdealGPT(ours)** | **50.7** |

Table 1: Accuracy of VCR Q→A task (ZS: Zero-Shot, Sup: Supervised)

|  |  | Acc.(%) |
|---|---|---|
|  | Random Guess | 33.3 |
| Sup. | EVE-Image (Xie et al., 2019) | 71.6 |
| Sup. | UNITER (Chen et al., 2020) | 79.4 |
| Sup. | OFA (Wang et al., 2022a) | 91.0 |
| ZS. | MiniGPT4 (Zhu et al., 2023b) | 35.1 |
| ZS. | LLaVA (Liu et al., 2023) | 40.3 |
| ZS. | **IdealGPT(ours)** | **55.3** |

Table 2: Accuracy of SNLI-VE (ZS: Zero-Shot, Sup: Supervised)

used to distinguish it from others. However, most existing VLMs cannot understand bounding boxes in the text input, making them hard to perform VCR directly. To alleviate that, we conduct a pre-processing to describe the mentioned person's spatial coordinate in words easy-to-understand. Please see details in Appendix E.

Although a lot of models can be finetuned on this task (Zellers et al., 2019; Li et al., 2019), there is hardly any model that tackles it in the zero-shot fashion. We empirically tried BLIP-2, MiniGPT4, and LLaVA and found that only the GPT4-like models can functionally perform zero-shot VCR, while other models such as BLIP-2 fail to understand the long context and the instruction of finding the correct answer. We present the experimental result in Tab. 1, where all zero-shot results are obtained from the randomly sampled 5000 data in the validation set. As we can see, IdealGPT can outperform the best GPT4-like model, MiniGPT4 by over 10%. It's noted that in IdealGPT reported here, BLIP2-FlanT5-XL is used as the Answerer. Please refer to Sec. 4.4 for ablations on the choice of Answerer.

In Fig.2, we showcase an example of how Ideal-GPT solves an example in VCR successfully. As we can see, in the first pass, the generated sub-questions and predicted sub-answers are not informative enough to support a solid conclusion because their identities and interaction of "watching"" cannot quite indicate whether they are dating or not. Further, in the second pass, after inputting the analysis from the Reasoner and existing sub-questions and sub-answers, the Questioner is prompted to ask additional supplementary sub-questions to collect more evidence about their expressions and body language. As a result, the updated sub-

answers of sad expressions and distant body language allow the Reasoner to ensure a confident final answer.

### 4.3 Visual Entailment

Visual Entailment requires the model to predict whether the image semantically entails the text. In each sample, there is a pair of an image and a text hypothesis along with three answer candidates (*i.e.*, entailment, neutral, and contradiction). The model needs to select one of them to represent the relationship between the image and the text hypothesis. It is challenging to solve the three-category classification task in a zero-shot manner. As shown in Fig.3, to begin with, we make some rules and introduce the goal of this task to ChatGPT[1] to ensure it can understand our instructions and respond reasonably. We utilize VLMs to generate the image caption and inject it into the prompt. Thereafter, Questioner decomposes the original hypothesis into several sub-questions. After answering all sub-questions by VLMs, Reasoner will summarize the image caption, hypothesis, all sub-questions, and corresponding sub-answers together to provide a comprehensive analysis. In Fig.3, we should notice that not only does IdealGPT exhibit the perceptional ability in sub-question 1 and 2, but also it shows the strong commonsense reasoning ability in sub-question 3.

Supervised methods for SNLI-VE have been well-studied (Xie et al., 2019; Chen et al., 2020; Wang et al., 2022a), while zero-shot approaches are less explored. We tried LLaVA and MiniGPT4 to do zero-shot SNLI-VE. In Tab. 2, we can observe that compared to MiniGPT4 and LLaVA, IdealGPT consistently surpasses them by a large margin, 20%, and 15% respectively. This result shows that not only can our method understand long instructions

| Model | Max. #Iterations | Acc.(%) |
|---|---|---|
| | 1 | 49.2 |
| | 2 | 53.2 |
| IdealGPT | 4 | 55.8 |
| | 6 | 57.3 |
| | 7 | 57.4 |

Table 3: Ablation of iterative decomposing. Max. #Iterations=1 means deterministic answering in one round without iterative decomposing.

| Model | Answerer | Acc.(%) |
|---|---|---|
| | BLIP-2 | 55.8 |
| IdealGPT | MiniGPT4 | 52.8 |
| | LLaVA | 53.2 |

Table 4: Ablation of choice of `Answerer`.

| Model | Caption from | Acc.(%) |
|---|---|---|
| | BLIP-2 | 55.8 |
| IdealGPT | MiniGPT4 | 48.4 |
| | LLaVA | 51.2 |

Table 5: Ablation of generated captions.

but also it is able to handle different task formats (SNLI-VE and VCR have distinctively different task formats. The former is image-hypothesis pair but the latter is the question-answering format). From Tab. 2, we can also notice that the performance of MiniGPT4 is the near random-guessing level. More details and discussion about SNLI-VE can be found in Appendix B.

## 4.4 Ablation Studies

In this section, we ablate both the design choice and component choice. We first demonstrate the necessity of iterative design in IdealGPT. Then we ablate different VLMs for generating captions and performing as `Answerer`. In all ablations, we use a randomly sampled 500 data set from VCR, which in our findings is enough to distinguish different model choices.

**Iterative Decomposing.** A key design in IdealGPT is that if the model (`Reasoner`) is not sure about the final answer, it will trigger a new pass of QA to ask additional sub-questions and therefore provide more visual facts to the `Reasoner`. The iterative decomposing will continue until the Reasoner is confident to make a decision or the number of passes reaches a pre-defined bound, which is a hyperparameter. We evaluate IdealGPT with iterative decomposing and without iterative decomposing (*i.e.*, Max.#Iterations=1) in Tab. 3. We can see that iterative decomposing design can boost the accuracy by as high as around 8%. It's noted that more passes also mean more inference time, and we find setting the maximum number of iterations to 4 achieves a good trade-off between efficiency and effectiveness, which is used as default in all other experiments. Under the above setting, the average number of passes across sampled data is 1.8.

**Answerer Choice.** As we mentioned before, `Answerer` can be any VLM capable of answering

visual questions. We mainly ablate three VLMs: BLIP-2, MiniGPT4, and LLaVA. The result is shown in Tab. 4. Although MiniGPT4 and LLaVA can follow instructions and understand the longer contexts, they are worse than BLIP-2 when answering questions related to detailed visual facts. This also echoes the limitation mentioned in their papers about hallucinations and the lack of spatial/geometric understanding. Note that the BLIP-2 we use is BLIP2-FlanT5-XL, which in our experiments, gives a similar performance as BLIP2-FlanT5-XXL.

**Image Captions.** To efficiently search for the best caption in our method, we fix the VLM as BLIP2-FlanT5-XL and go through pre-trained VLMs without fine-tuning on any caption dataset. From the experimental results shown in Tab. 5, we see BLIP-2 exhibits the best performance. It is interesting to observe that both MiniGPT4 and LLaVA have shown impressive captioning ability in their papers/demos, but they fail in our framework when compared with BLIP-2. We further go through many captions generated from MiniGPT4/LLaVA and BLIP-2 and find that MiniGPT4/LLaVA tends to generate longer, more informative but less accurate (or more hallucination) captions. In contrast, BLIP-2 tends to generate short, less informative, but also less mistaken captions. So the hallucinations/mistakes in MiniGPT4/LLaVA-generated captions tend to mislead the `Questioner` and `Reasoner` to wrong answers.

## 4.5 Error Analysis

Since our method consists of multiple components, it would be interesting to see where the error comes from. We went through 50 failure samples of Ideal-

| Source | Type | Ratio.(%) |
|---|---|---|
| Questioner | Not Relevant Sub-questions | 16 |
| Answerer | Wrong Sub-answers | 52 |
| Reasoner | Hallucination | 24 |
| | Misunderstanding | 8 |

Table 6: Error Analysis of different components

GPT on VCR validation set and conducted an error analysis about different types of errors from different components, thanks to the good transparency and interpretability of IdealGPT. The quantitative result is shown in Tab. 6. Below are more detailed analyses.

Questioner: We found that the Questioner sometimes generates sub-questions that are not quite relevant to the four answer choices, thus not helpful in distinguishing the correct answer. Besides the capability of LLM itself, we found it's also heavily influenced by the caption we provided. If the caption is more precise with less hallucination, then the Questioner would be more precise too.

Answerer: Half of the errors come from the Answerer providing wrong sub-answers. As illustrated in Tab. 4, we can find that the choice of Answerer is quite important. A good Answerer should understand various types of questions and generate answers with less hallucination.

Reasoner: There are two types of errors from Reasoner: a). Hallucination: Sometimes Reasoner predicts the answer only with unclear visual clues, thus leading to wrong prediction. In those cases, one more round of the decomposition loop might help to find more clear visual evidence. This also accords with our observation that more decomposition iterations help. b). Misunderstanding: Sometimes, the LLM misunderstands the answer candidates and gives wrong predictions, which is mainly due to the LLM's capacity.

## 5  Conclusion

In this work, we identify critical problems with existing VQA methods, especially the lack of addressing zero-shot reasoning tasks and the false assumption potentially forcing models to answer questions without sufficient information. To address these issues, we propose IdealGPT to utilize LLMs to construct a multiple-passes framework among a Questioner, a Answerer, and a Reasoner. This framework ensures better interpretability of VLMs' reasoning and robustness to re-correctifying predictions such as hallucination. Additionally, the generalizability of our framework can be illustrated by the modularity and our superior zero-shot performance. Our extensive experiments prove the effectiveness of each component in IdealGPT and verify that it can outperform the best existing GPT-4-like models by an absolute 10% in VCR and 15% in SNLI-VE.

## Acknowledgements

This work is supported by DARPA MCS program under Cooperative Agreement N66001-19-2-4032.

## Limitations

Although we have witnessed the strong performance of the proposed method, we still have some limitations. Firstly, answering key sub-questions correctly plays a significant role in the success of our system. Therefore, our final results are largely bottlenecked by the performance of pre-trained VLMs. Additionally, we just employ the general image caption to give ChatGPT an idea of what the image looks like. However, dense captioning might be more informative and better help ChatGPT deal with the unseen image. Besides, the prompts are manually designed by us and it is difficult to find the optimal prompt in a specific situation. It will be better if the prompt can be generated automatically. Last but not least, compared with end-to-end solutions, ours naturally takes more time and has higher latency because of multiple feedforward passes of Questioner/Answerer/Reasoner.

## Ethics Statement

ChatGPT is pre-trained on the colossal corpus which is likely to contain potential racial and gender bias. Therefore, if someone finds our work interesting and would like to use it in a specific environment, we strongly suggest the user check the potential bias before usage. In addition, it is hard to control the generation of LLMs like ChatGPT. We should be aware of the potential problems caused by hallucinations.

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

## A  Data Statistics

| VL Tasks | | Number of Samples |
|---|---|---|
| SNLI-VE | C | 1664 |
| | N | 1672 |
| | E | 1664 |
| | All | 5000 |
| VCR | - | 5000 |

Table 7: Statistics of SNLI-VE and VCR (C: Contradiction, N: Neural, E: Entailment, VL: Vision-Language)

We randomly select 5000 VCR and SNLI-VE samples from respective val and dev split. Zero-shot learning is conducted on these samples.

## B  Details of Visual Entailment Results

| | | C | N | E | All |
|---|---|---|---|---|---|
| | Random Guess | - | - | - | 33.3 |
| Sup. | EVE-Image (Xie et al., 2019) | 71.0 | 70.6 | 73.1 | 71.6 |
| | UNITER (Chen et al., 2020) | - | - | - | 79.4 |
| | OFA (Wang et al., 2022a) | - | - | - | 91.0 |
| ZS. | MiniGPT4 (Zhu et al., 2023b) | 3.2 | 3.2 | **99.0** | 35.1 |
| | LLaVA (Liu et al., 2023) | 12.0 | **31.3** | 77.6 | 40.3 |
| | **IdealGPT(ours)** | **83.4** | 25.9 | 56.7 | **55.3** |

Table 8: Accuracy of SNLI-VE (ZS: Zero-Shot, Sup: Supervised, C: Contradiction, N: Neutral, E: Entailment)

From the results shown in Tab.8, we can observe that our proposed method outperforms MiniGPT4 and LLaVA by a large margin. Moreover, MiniGPT4 exhibits near-chance level performance. When it is faced with different samples, it always replies *entailment*, which indicates that it doesn't obtain a strong reasoning ability to process and understand the visual entailment task. However, IdealGPT can achieve strong zero-shot results and even surpass the supervised EVE-Image model in *contradiction* category.

## C  Augmented Outside Knowledge Visual Question Answering

A-OKVQA is a challenging benchmark for knowledge-required visual question answering, which demands world knowledge that goes beyond the image. It provides Multiple-Choice (MC) as well as Direct Answer (DA) evaluation settings.

| | | Acc.(%) |
|---|---|---|
| | Random Guess | 25.0 |
| Sup. | LXMERT (Tan and Bansal, 2019) | 51.4 |
| | GPV-2 (Kamath et al., 2022) | 60.3 |
| | InstructBLIP (Dai et al., 2023) | 81.0 |
| ZS. | MiniGPT4 (Zhu et al., 2023b) | 49.4 |
| | LLaVA (Liu et al., 2023) | 30.0 |
| | **IdealGPT(ours)** | **62.6** |

Table 9: Accuracy of A-OKVQA in Multiple-Choice setting(ZS: Zero-Shot, Sup: Supervised)

Since it is difficult to evaluate the generated answer in the open-vocabulary setting, we choose the MC evaluation setting. We design the prompt for A-OKVQA[2] and directly input the question, four answer choices, and the general image caption into the `Questioner` to put forward several sub-questions. Afterward, the VLMs can reply to the `Questioner` and `Reasoner` is able to collect and analyze the sub-questions and sub-answers to output a prediction when it is sure about its answer. In Tab. 9, we investigate zero-shot A-OKVQA by using MiniGPT4 and LLaVA. IdealGPT can surpass MiniGPT4 and LLaVA by 13.2 % and 32.6 % respectively, which confirms the stronger reasoning ability of our proposed method. Moreover, we outperform some early proposed supervised methods like LXMERT (Tan and Bansal, 2019) and GPV-2 (Kamath et al., 2022). Since we implement our method without any further training, we still can't exceed InstructBLIP (Dai et al., 2023), which is one of the current best-performing supervised models on A-OKVQA.

## D  Details of Prompts Used

The prompts of IdealGPT used in the VCR task are shown in Fig. 4. As for SNLI-VE, the prompts are shown in Fig. 5. It's noted that the [placeholder] means we will replace it with the corresponding text of the instance, such as a main question, caption, and four choices.

## E  VCR Pre-Processing

We exploit two different approaches to pre-process VCR and select the better one for our experiments in the main text. The first pre-processing is to divide the image region from left to right into three

---

[2]Prompts for A-OKVQA are very similar to VCR in Appendix D

|  |  | Acc.(%) |
|---|---|---|
| **TB.** | MiniGPT4 (Zhu et al., 2023b) | 42.4 |
|  | LLaVA (Liu et al., 2023) | 31.0 |
| **DB.** | MiniGPT4 (Zhu et al., 2023b) | 30.8 |
|  | LLaVA (Liu et al., 2023) | 28.6 |

Table 10: Comparison of two different pre-processing ways for VCR (TB: Three Bins, DB: Drawn-on Boxes)

bins and check if the mentioned object's center point belongs to which bin. If it's in the most left bin, it's renamed as "person on the left". Similarly, "person in the middle" is in the middle bin, and "person on the right" is in the right bin. Since most QAs in VCR mentioned less than three persons, it can cover most cases. The second approach is to follow past work (Zellers et al., 2022) in 'drawing on' the annotated detection tags to the image. We select seven different colors (*i.e.*, red, green, blue, orange, purple, cyan, and yellow) to represent the different persons mentioned in the question or answer. Assume there are two people in one sample, when doing the inference, 'person1' will be replaced by 'person in the red bounding box', and 'person2' will be replaced by 'person in the green bounding box'. Moreover, the model can see the red and green bounding boxes drawn in the image. As we mentioned above, most cases in VCR mentioned less than three people. Therefore, our implementation can cover the majority of the cases. The comparison of these two methods can be found in Tab. 10. We randomly select 500 samples from VCR val split to conduct zero-shot learning by using LLaVA and MiniGPT4 to see which method is better. We can observe that the former approach is better, so we utilize this setting in the main text.

**VCR Questioner Prompt:**

---

**System Prompt of 1st Iteration:**

You are an AI assistant who has rich visual commonsense knowledge and strong reasoning abilities.
You will be provided with:
1. A main question about an image and four answer candidates.
2. Although you won't be able to directly view the image, you will receive a general caption that might not be entirely precise but will provide an overall description.

Your goal is:
To effectively analyze the image and select the correct answer for the question, you should break down the main question into several sub-questions that address the key aspects of the image.

Here are the rules you should follow when listing the sub-questions.
1. Ensure that each sub-question is independent. It means the latter sub-questions shouldn't mention previous sub-questions.
2. List the sub-questions in the following format: "Sub-question 1: ...?; Sub-question 2: ...?".
3. Each sub-question should start with "What".
4. Each sub-question should be short and easy to understand.
5. The sub-question are necessary to distinguish the correct answer.

Example:
Main question: What is happening in the image?
Sub-question 1: What objects or subjects are present in the image?
Sub-question 2: What actions or events is the person doing?
Sub-question 3: What are the emotions or expressions of the woman?
Sub-question 4: What is the brand of this car?

---

**System Prompt of Following Iterations:**

You are an AI assistant who has rich visual commonsense knowledge and strong reasoning abilities.
You will be provided with:
1. A main question about an image and four answer candidates.
2. Although you won't be able to directly view the image, you will receive a general caption that might not be entirely precise but will provide an overall description.
3. Some sub-questions decomposed from the main question, and the corresponding answers are provided by a visual AI model. It's noted that the answers are not entirely precise.
4. An analysis of whether the given sub-questions and sub-answers can help to solve the original main question.

The current sub-questions and sub-answers are not sufficient to solve the main question. Your goal is:
Based on existing sub-questions and analysis, you should pose additional questions, that can gather more information and are necessary to solve the main question.

Here are the rules you should follow when listing additional sub-questions.
1. Ensure that each sub-question is independent. It means the latter sub-questions shouldn't mention previous sub-questions.
2. List the sub-questions in the following format: "Additional Sub-question 1: ...?; Additional Sub-question 2: ...?".
3. Each sub-question should start with "What".
4. Each sub-question should be short and easy to understand.
5. The sub-question are necessary to distinguish the correct answer.

Format Example:

Additional Sub-question 1: xxxx
Additional Sub-question 2: xxxx
Additional Sub-question 3: xxxx
Additional Sub-question 4: xxxx

---

**Input Prompt of 1st Iteration:**

Imperfect Caption: [placeholder]
Main Question: [placeholder]
Four Choices: [placeholder]
Please list the sub-questions following the requirement I mentioned before.

---

**Input Prompt of Following Iterations:**

Imperfect Caption: [placeholder]
Main Question: [placeholder]
Four Choices: [placeholder]
Sub-questions and answers: [placeholder]
Analysis: [placeholder]
Please list the sub-questions following the requirement I mentioned before.

---

**VCR Answerer Prompt:**

Question: [placeholder] Answer:

---

**VCR Reasoner Prompt:**

---

**System Prompt of All but Last Iteration:**

You are an AI assistant who has rich visual commonsense knowledge and strong reasoning abilities.
You will be provided with:
1. A main question about an image and four answer candidates.
2. Although you won't be able to directly view the image, you will receive a general caption that might not be entirely precise but will provide an overall description.
3. Some sub-questions decomposed from main question, and the corresponding answers are provided by a visual AI model. It's noted that the answers are not entirely precise.

Your goal is:
Based on sub-questions and corresponding answers, you should find the more likely answer from the four answer candidates.

Here are the rules you should follow in your response:
1. At first, demonstrate your reasoning and inference process within one paragraph. Start with the format of "Analysis:".
2. If you have found the more likely answer, conclude the correct answer id in the format of "More Likely Answer: 1/2/3/4". Otherwise, conclude with "More Likely Answer: We are not sure which option is correct".

Response Format:
Analysis: xxxxxx.
More Likely Answer: 1/2/3/4.

---

**System Prompt of Last Iteration:**

You are an AI assistant who has rich visual commonsense knowledge and strong reasoning abilities.
You will be provided with:
1. A main question about an image and four answer candidates.
2. Although you won't be able to directly view the image, you will receive a general caption that might not be entirely precise but will provide an overall description.
3. Some sub-questions decomposed from main question, and the corresponding answers are provided by a visual AI model. It's noted that the answers are not entirely precise.

Your goal is:
Based on sub-questions and corresponding answers, you must find the more likely answer from the four answer candidates.

Here are the rules you should follow in your response:
1. At first, demonstrate your reasoning and inference process within one paragraph. Start with the format of "Analysis:".
2. Tell me the more likely answer's id in the format of "More Likely Answer: 1/2/3/4". Even if you are not confident, you must give a prediction with educated guessing.

Response Format:
Analysis: xxxxxx.
More Likely Answer: 1/2/3/4.

---

**Input Prompt:**

Imperfect Caption: [placeholder]
Main Question: [placeholder]
Four Choices: [placeholder]
Existing Sub-questions and answers: [placeholder]
Please follow the above-mentioned instruction to list the Analysis and More Likely Answer.

Figure 4: The prompts of IdealGPT in VCR task.

**SNLI-VE Questioner Prompt:**

**System Prompt of 1st Iteration:**

You are an AI assistant who has rich visual commonsense knowledge and strong reasoning abilities.
You will be provided with:
1. A textual hypothesis about an image and three answer candidates.
2. Although you won't be able to directly view the image, you will receive a general caption that might not be entirely precise but will provide an overall description.

Your goal is:
To effectively predict whether the image semantically entails the textual hypothesis and select the answer from entailment, neutral, and contradiction, you should come up with several sub-questions that address the key aspects of the image.

Here are the rules you should follow when listing the sub-questions.
1. Ensure that each sub-question is independent. It means the latter sub-questions shouldn't mention previous sub-questions.
2. List the sub-questions in the following format: "Sub-question 1: ...?; Sub-question 2: ...?".
3. Each sub-question should start with "What".
4. Each sub-question should be short and easy to understand.
5. The sub-questions are necessary to distinguish the correct answer.

Example:

Hypothesis: A group of women are walking along the railroad tracks.
Sub-question 1: What objects or subjects are present in the image?
Sub-question 2: What actions or events are the people doing?
Sub-question 3: What is the location where the people are walking?
Sub-question 4: What is the gender of this group of people?

**System Prompt of Following Iterations:**

You are an AI assistant who has rich visual commonsense knowledge and strong reasoning abilities.
You will be provided with:
1. A textual hypothesis about an image and three answer candidates.
2. Although you won't be able to directly view the image, you will receive a general caption that might not be entirely precise but will provide an overall description.
3. Some sub-questions proposed for predicting whether the image semantically entails the textual hypothesis, and the corresponding answers are provided by a visual AI model. It's noted that the answers are not entirely precise.
4. An analysis of whether the given sub-questions and sub-answers can help to predict whether the image semantically entails the textual hypothesis.

The current sub-questions and sub-answers are not sufficient to predict whether the image semantically entails the textual hypothesis. Your goal is:
Based on existing sub-questions and analysis, you should pose additional questions, that can gather more information and are necessary to predict whether the image semantically entails the textual hypothesis.

Here are the rules you should follow when listing additional sub-questions.
1. Ensure that each sub-question is independent. It means the latter sub-questions shouldn't mention previous sub-questions.
2. List the sub-questions in the following format: "Additional Sub-question 1: ...?; Additional Sub-question 2: ...?".
3. Each sub-question should start with "What".
4. Each sub-question should be short and easy to understand.
5. The sub-questions are necessary to distinguish the correct answer.

Format Example:

Additional Sub-question 1: xxxx
Additional Sub-question 2: xxxx
Additional Sub-question 3: xxxx
Additional Sub-question 4: xxxx

**Input Prompt of 1st Iteration:**

Imperfect Caption: [placeholder]
Hypothesis: [placeholder]
Three Choices: [placeholder]
Please list the sub-questions following the requirement I mentioned before.

**Input Prompt of Following Iterations:**

Imperfect Caption: [placeholder]
Hypothesis: [placeholder]
Three Choices: [placeholder]
Sub-questions and answers: [placeholder]
Analysis: [placeholder]
Please list the sub-questions following the requirement I mentioned before.

**SNLI-VE Answerer Prompt:**

Question: [placeholder] Answer:

**SNLI-VE Reasoner Prompt:**

**System Prompt of All but Last Iteration:**

You are an AI assistant who has rich visual commonsense knowledge and strong reasoning abilities.
You will be provided with:
1. A textual hypothesis about an image and three answer candidates.
2. Although you won't be able to directly view the image, you will receive a general caption that might not be entirely precise but will provide an overall description.
3. Some sub-questions proposed for predicting whether the image semantically entails the textual hypothesis, and the corresponding answers are provided by a visual AI model. It's noted that the answers are not entirely precise.

Your goal is:
Based on sub-questions and corresponding answers, you should find the more likely answer from the three answer candidates.

Here are the rules you should follow in your response:
1. At first, demonstrate your reasoning and inference process within one paragraph. Start with the format of "Analysis:".
2. If you have found the more likely answer, conclude the correct answer in the format of "More Likely Answer: entailment/neutral/contradiction". Otherwise, conclude with "More Likely Answer: We are not sure which option is correct".

Response Format:
Analysis: xxxxxx.
More Likely Answer: entailment/neutral/contradiction.

**System Prompt of Last Iteration:**

You are an AI assistant who has rich visual commonsense knowledge and strong reasoning abilities.
You will be provided with:
1. A textual hypothesis about an image and three answer candidates.
2. Although you won't be able to directly view the image, you will receive a general caption that might not be entirely precise but will provide an overall description.
3. Some sub-questions proposed for predicting whether the image semantically entails the textual hypothesis, and the corresponding answers are provided by a visual AI model. It's noted that the answers are not entirely precise.

Your goal is:
Based on sub-questions and corresponding answers, you must find the more likely answer from the three answer candidates.

Here are the rules you should follow in your response:
1. At first, demonstrate your reasoning and inference process within one paragraph. Start with the format of "Analysis:".
2. Tell me the more likely answer in the format of "More Likely Answer: entailment/neutral/contradiction". Even if you are not confident, you must give a prediction with educated guessing.

Response Format:
Analysis: xxxxxx.
More Likely Answer: entailment/neutral/contradiction.

**Input Prompt:**

Imperfect Caption: [placeholder]
Hypothesis: [placeholder]
Three Choices: [placeholder]
Existing Sub-questions and answers: [placeholder]
Please follow the above-mentioned instruction to list the Analysis and More Likely Answer.

Figure 5: The prompts of IdealGPT in SNLI-VE task.