# OpenReview forum: "IdealGPT: Iteratively Decomposing Vision and Language Reasoning via Large Language Models"
_EMNLP/2023/Conference — EMNLP 2023 Findings_

### Official Review · Reviewer_duK1 · 2023-07-31

**Soundness:** 3

**Excitement:**

3: Ambivalent: It has merits (e.g., it reports state-of-the-art results, the idea is nice), but there are key weaknesses (e.g., it describes incremental work), and it can significantly benefit from another round of revision. However, I won't object to accepting it if my co-reviewers champion it.

**Paper Topic And Main Contributions:**

This paper introduces a zero-shot framework, IdealGPT, on open-domain visual question-answering task. Specifically, the framework IdealGPT iteratively decomposes VL reasoning using large language models. The framework has three components, a questioner, an answer and a reasoner. Given an image and a question, the reasoner checks whether it is confident to answer the question. If not, the questioner asks more questions and the answer answers them. The reasoner will use new information to answer the original question again.

**Questions For The Authors:**

1. What types of the errors did the model make? Are the errors from the reasoner, questioner or answerer?
2. If using different VLMs, what would be the effects? Does this method heavily depend on the capabilities of VLMs?
3. For the ZS. performance, the baseline models, such as MiniGPT4 or LLaVA have very low performance. Why do these baseline models have bad performance on these tasks? Did you carefully work on the prompt engineering for LLaVA or MiniGPT4?
4. In VCR dataset, some questions will use [obj1] or [obj2] to indicate the objects in the image. How did you process this in the IdealGPT framework?
5. Does the LLM ask wrong or irrelevant questions?
6. I realize that some baselines such as LLaVA have very low performance. One of the reason might be that under zero-shot setting, LLaVA does not have strong context learning ability. What would be the performance if the chatGPT directly answer the questions?
7. Have you compared your results with ChatCaptioner as you mentioned in the related work? Also, some of the works use the similar tricks and we might want to compare with them, such as GPT + CoT.

**Reasons To Accept:**

The framework is novel and easy to reproduce. It could be useful for various vision-language tasks.

**Reasons To Reject:**

The framework is a zero-shot setting and the performance is influenced significantly by the capabilities of the VLMs and the LLMs. The paper did not explore too much on how different VLM could influence the performance. Also, the paper lacks some essential baselines such that the performance improvements are not convinced.

**Reproducibility:**

5: Could easily reproduce the results.

**Reviewer Confidence:**

4: Quite sure. I tried to check the important points carefully. It's unlikely, though conceivable, that I missed something that should affect my ratings.

---

### Official Review · Reviewer_NAya · 2023-08-05

**Soundness:** 3

**Excitement:**

4: Strong: This paper deepens the understanding of some phenomenon or lowers the barriers to an existing research direction.

**Paper Topic And Main Contributions:**

This paper addresses the problem of zero-shot vision-and-language (VL) reasoning tasks that require multi-step inferencing. The main contribution is the proposed IdealGPT framework, which iteratively decomposes VL reasoning using large language models (LLMs). IdealGPT consists of three modules: a Questioner (LLM) to generate sub-questions, a pre-trained VL model (VLM) as the Answerer to provide corresponding sub-answers, and another LLM as the Reasoner to infer the final answer. These modules perform the divide-and-conquer procedure iteratively until the model is confident about the final answer to the main question. IdealGPT is evaluated on multiple challenging VL reasoning tasks under a zero-shot setting, outperforming the best existing GPT-4-like models by an absolute 10% on VCR and 15% on SNLI-VE.

**Questions For The Authors:**

Question A: How does the performance of IdealGPT change when using different Answerers, especially when using VLMs with different levels of performance or specific strengths and weaknesses?

Question B: Can you provide more insights into the trade-offs between the number of iterations and the accuracy of IdealGPT? For example, how does the performance change with more than 4 iterations, and is there a point of diminishing returns?

Question C: Have you considered exploring automatic prompt generation or optimization techniques to improve the performance of IdealGPT without relying on manually designed prompts?

Question D: Are there any strategies or methods to mitigate the potential biases and hallucination issues introduced by ChatGPT when used as the Questioner and Reasoner in IdealGPT?

**Reasons To Accept:**

1. IdealGPT enables a clear understanding of the reasoning process by decomposing the main question into sub-questions and sub-answers, making it easier to pinpoint any issues in the model's predictions.
2. The framework can be easily updated with more powerful LLMs and VLMs to improve its performance, as it separates the components of Questioner, Answerer, and Reasoner.
3. Generalizability: IdealGPT can be seamlessly applied to various tasks by slightly adjusting the prompt and demonstrates strong zero-shot ability without training or fine-tuning on specific tasks.
4. Empirical results showing that IdealGPT outperforms state-of-the-art GPT-4-like models in zero-shot settings on challenging VL reasoning tasks, providing a new benchmark for future research.

**Reasons To Reject:**

1. The success of IdealGPT relies heavily on the Answerer's ability to correctly answer key sub-questions. The final results are limited by the performance of the pre-trained VLMs used as Answerers.
2. IdealGPT requires multiple feedforward passes for the Questioner, Answerer, and Reasoner, which may increase latency and computational costs compared to end-to-end solutions.
3. The use of ChatGPT as the Questioner and Reasoner might introduce biases or hallucinations present in the pre-trained LLM, affecting the overall reasoning process.
4. The prompts for the LLMs are manually designed, making it challenging to find the optimal prompt for a specific situation.
5. The dependency on the performance of the Answerer (VLM) may limit the generalizability of the framework when dealing with different tasks or situations where the pre-trained VLMs perform poorly.

**Reproducibility:**

4: Could mostly reproduce the results, but there may be some variation because of sample variance or minor variations in their interpretation of the protocol or method.

**Reviewer Confidence:**

4: Quite sure. I tried to check the important points carefully. It's unlikely, though conceivable, that I missed something that should affect my ratings.

---

### Official Review · Reviewer_MWQz · 2023-08-07

**Soundness:** 3

**Excitement:**

3: Ambivalent: It has merits (e.g., it reports state-of-the-art results, the idea is nice), but there are key weaknesses (e.g., it describes incremental work), and it can significantly benefit from another round of revision. However, I won't object to accepting it if my co-reviewers champion it.

**Paper Topic And Main Contributions:**

The paper proposed IdealGPT to utilize LLMs to construct a multiple-passes framework for zero-shot VQA tasks.
The proposed method outperforms other zero-shot methods.

**Reasons To Accept:**

1. The proposed method surpasses other methods on VCR and SNLI-VE tasks.
2. This paper is well written and easy to follow.

**Reasons To Reject:**

1. Comparative methods are limited. The authors should include other baselines (such as VQG [Uehara et al., 2022], Co-VQA [Wang et al., 2022] that use sub-questions to predict final answers to evaluate the performance of the proposed method even though they are not zero-shot methods.
2. More evaluation datasets are required. The proposed method should be also tested on other tasks such as visual question answering or OK-VQA to show generalizability of the method.

**Reproducibility:**

2: Would be hard pressed to reproduce the results. The contribution depends on data that are simply not available outside the author's institution or consortium; not enough details are provided.

**Reviewer Confidence:**

3: Pretty sure, but there's a chance I missed something. Although I have a good feel for this area in general, I did not carefully check the paper's details, e.g., the math, experimental design, or novelty.

---

### Meta-Review · Area_Chair_JUYc · 2023-09-19

**Recommendation:** 3

**Metareview:**

This paper proposes a new model called IdealGPT to utilize LLMs to construct a multiple inference steps for zero-shot VQA tasks. The proposed method outperforms other zero-shot methods.

The reviewers were all in agreement that the method is sound with good experimental studies, strong results, and interesting ideas. Most of the concerns regarding soundness were minor, regarding more discussion of complexity, ablation studies, experimental details, and also asking for more comparisons and datasets regarding different LLM reasoning abilities, risk of hallucination, and choice of prompts. The reviewers adequately addressed these concerns during the discussion period. Overall, the reviewers were mixed in their excitement regarding the paper, with 2 ambivalent and 1 finding it strong. The reviewers who were ambivalent largely found that the proposed method mostly inherits known limitations of language models, which might limit its use due to existing issues wrt reasoning abilities, risk of hallucination, and sensitivity to prompts.

---

### Decision · Program_Chairs · 2023-10-07

**Decision:**

Accept-Findings

**Comment:**

This paper proposes a new model called IdealGPT to utilize LLMs to construct a multiple inference steps for zero-shot VQA tasks. The proposed method outperforms other zero-shot methods.

The reviewers were all in agreement that the method is sound with good experimental studies, strong results, and interesting ideas. Most of the concerns regarding soundness were minor, regarding more discussion of complexity, ablation studies, experimental details, and also asking for more comparisons and datasets regarding different LLM reasoning abilities, risk of hallucination, and choice of prompts. The reviewers adequately addressed these concerns during the discussion period. Overall, the reviewers were mixed in their excitement regarding the paper, with 2 ambivalent and 1 finding it strong. The reviewers who were ambivalent largely found that the proposed method mostly inherits known limitations of language models, which might limit its use due to existing issues wrt reasoning abilities, risk of hallucination, and sensitivity to prompts.